# Gastric Cancer Angiogenesis Assessment by Dynamic Contrast Harmonic Imaging Endoscopic Ultrasound (CHI-EUS) and Immunohistochemical Analysis—A Feasibility Study

**DOI:** 10.3390/jpm12071020

**Published:** 2022-06-21

**Authors:** Victor Mihai Sacerdoțianu, Bogdan Silviu Ungureanu, Sevastiţa Iordache, Sergiu Marian Cazacu, Daniel Pirici, Ilona Mihaela Liliac, Daniela Elena Burtea, Valeriu Șurlin, Cezar Stroescu, Dan Ionuț Gheonea, Adrian Săftoiu

**Affiliations:** 1Research Center of Gastroenterology and Hepatology of Craiova, University of Medicine and Pharmacy of Craiova, 200349 Craiova, Romania; sacerdotianumihai@gmail.com (V.M.S.); sevastita@gmail.com (S.I.); cazacu2sergiu@yahoo.com (S.M.C.); dana.burtea26@gmail.com (D.E.B.); digheonea@gmail.com (D.I.G.); adrian.saftoiu@umfcv.ro (A.S.); 2Histology Department, University of Medicine and Pharmacy of Craiova, 200349 Craiova, Romania; daniel.pirici@umfcv.ro (D.P.); ilona.mihaela.liliac@gmail.com (I.M.L.); 3Surgical Department, University of Medicine and Pharmacy of Craiova, 200349 Craiova, Romania; vsurlin@gmail.com; 4Surgical Department II, St. Mary Hospital Bucharest, 011172 București, Romania; cezar.stroescu@gmail.com

**Keywords:** CHI-EUS, angiogenesis, gastric cancer

## Abstract

Tumor vascular perfusion pattern in gastric cancer (GC) may be an important prognostic factor with therapeutic implications. Non-invasive methods such as dynamic contrast harmonic imaging endoscopic ultrasound (CHI-EUS) may provide details about tumor perfusion and could also lay out another perspective for angiogenesis assessment. Methods: We included 34 patients with GC, adenocarcinoma, with CHI-EUS examinations that were performed before any treatment decision. We analyzed eighty video sequences with a dedicated software for quantitative analysis of the vascular patterns of specific regions of interest (ROI). As a result, time-intensity curve (TIC) along with other derived parameters were automatically generated: peak enhancement (PE), rise time (RT), time to peak (TTP), wash-in perfusion index (WiPI), ROI area, and others. We performed CD105 and CD31 immunostaining to calculate the vascular diameter (vd) and the microvascular density (MVD), and the results were compared with CHI-EUS parameters. Results: High statistical correlations (*p* < 0.05) were observed between TIC analysis parameters MVD and vd CD31. Strong correlations were also found between tumor grade and 7 CHI-EUS parameters, *p* < 0.005. Conclusions: GC angiogenesis assessment by CHI-EUS is feasible and may be considered for future studies based on TIC analysis.

## 1. Introduction

Gastric cancer (GC) is one of the most common gastrointestinal malignancies, and despite its decreasing incidence in the past years, there are still more than 750,000 cases diagnosed annually worldwide. While curative options are available for patients diagnosed in early stages, the survival rate for advanced stages remains very poor, thus suggesting that available therapeutic options are suboptimal [1,2].

Angiogenesis is a critical process for tumor progression and has been considered one of the main therapeutic targets. The formation of new vessels seems to correlate with tumor progression, and GC makes no exception. While the VEGF pathway is one of the most studied angiogenetic processes, with a current monoclonal antibody directed against VEGFR-2, alternative pathways may occur and promote angiogenesis. Targeting tumor progression by focusing on newly developed tumor vessels might provide a better prognosis for patients. Currently, there are several FDA-approved therapies that are designed to target pro-angiogenic signaling in GC [3,4]; however, a proper assessment should be performed in order to validate the targeting method.

The common way to assess the neovascularization process is to measure the microvascular density (MVD) on immunohistochemical staining with several markers of vascular endothelial cells, such as factor VIII, CD31, CD34, and CD105 [5]. Endoglin (CD105), which promotes endothelial cell proliferation and migration, has a high specificity and sensibility for newly formed vessels. Moreover, it seems to be correlated with lymph node metastasis [6,7].

Dynamic functional imaging techniques have been proposed for vascular assessment and are currently used for diagnostic purposes in many cancers. Contrast-enhanced ultrasound methods have shifted from transabdominal ultrasound to endoscopic ultrasound (EUS) settings and have enhanced the diagnosis spectrum for different types of tumors. It is noteworthy that when discussing gastric tumors, contrast-enhanced endoscopic ultrasound has been used especially for the differential diagnosis of subepithelial tumors [8] and more recently [9] has been proposed for lymph-node assessment. Furthermore, dynamic contrast harmonic imaging endoscopic ultrasound (CHI-EUS) may assess the contrast agent-related signal in the tumoral area for an established period of time even from the low-velocity flow microvessels [10,11]. Dedicated software can quantitatively evaluate the gastric vascular pattern of a specific region of interest (ROI) and then automatically generate a diagram named time-intensity curve (TIC), with all derived parameters [12]. There are several studies on colorectal [13] and breast cancers [14] that found significant correlations between TIC parameters and MVD; however, to our knowledge, there are no similar studies on GC

Our objective was to evaluate the perfusion pattern in GC by providing a quantitative analysis of TIC parameters on CHI-EUS videos in comparison to immunohistochemical angiogenesis markers.

## 2. Materials and Methods

### 2.1. Patients

Fifty-six patients diagnosed with gastric tumors between 1 November 2019 to 1 January 2022 were included in our study. All patients were referred for EUS local staging at the Research Center of Gastroenterology and Hepatology of Craiova, University of Medicine and Pharmacy of Craiova, ROMANIA.

Inclusion criteria: (1) patients with endoscopic biopsy-proven diagnosis of gastric adenocarcinoma; (2) age from 18 to 90 years old; and (3) signed informed consent for endoscopic biopsies, EUS, and CHI-EUS examinations.

Exclusion criteria: (1) prior treatment with chemo-radiotherapy; (2) other histopathological types of gastric tumors (gastric lymphoma, gastrointestinal stromal tumors, neuroendocrine tumors); (3) gastroesophageal junction tumors; and (4) presence of health conditions that contraindicated deep sedation or EUS.

The study was approved by the ethics committee of University of medicine and pharmacy of Craiova, Romania (No. 09/12.01.2019), and all procedures were performed according to the Declaration of Helsinki after signing informed consent. This study was recorded on ClinicalTrials.gov, identifier: NCT05051423.

### 2.2. Contrast-Enhanced Harmonic Imaging Endoscopic Ultrasound

All patients were initially evaluated by EUS using a linear oblique-viewing echoendoscope (echoendoscope Olympus, GF-UCT180, tower Olympus Evis Exera II CV-180, Olympus Optical Corporation, Tokyo, Japan) coupled with ultrasound equipment Hitachi-Aloka Prosound Alpha 7, Hitachi Aloka Medical Ltd., Tokyo, Japan) able to perform harmonic imaging contrast examination. The procedures were performed by an experienced gastroenterologist in EUS. Biopsies were harvested from normal and pathologic tissue for angiogenesis immunomarkers validation. The tumors were characterized by describing their position, dimensions, echogenicity, echostructure, and depth invasion into the gastric wall with or without involving adjacent structures. Perigastric lymph nodes were suspected as malignant if the following criteria were met: low-level echoes and homogeneous, well-circumscribed, rounded, enlarged structures [15]. We used the TNM edition of the American Joint Committee on Cancer classification for GC staging [16]. Moreover, the tumoral presence or absence of a power Doppler signal was also noted.

After TNM EUS staging, the CHI-EUS examination was performed. A low mechanical index (dynamic wide-band contrast harmonic imaging mode) of 0.2 was chosen. The tumor was properly examined in conventional gray-scale B-mode until the desired area for examination was highlighted (Figure 1). For example, in the case of a tumor causing gastric outlet obstruction, only the proximal side of the lesion was assessed. An intravenous bolus injection of 4.8 mL of a second-generation contrast agent (SonoVue, Bracco, Milan, Italia) was injected followed by a 5 mL 0.9% sodium chloride flush. CHI-EUS examinations (T0-T120s) were assessed in real-time and recorded on a local HDD for later system analysis. The contrast enhancement pattern was noted.

### 2.3. Time-Intensity Curve Analysis

Quantitative analysis of CHI-EUS tumor perfusion pattern was evaluated using a post-processing platform with dedicated software entitled VueBox^®^ (Bracco Suisse SA, Plan-les-Ouates, Switzerland). CHI-EUS videos recorded were converted in DICOM format and processed using the aforementioned software. Four regions of interest (ROI) were highlighted inside the tumoral zone (Figure 2A). The TIC was analyzed, and the following parameters were automatically generated: peak enhancement (PE), the maximum intensity peak in TIC; wash-in area under the curve (WiAUC); rise time (RT), the time from the beginning of contrast enhancement to PE (Figure 2B); mean transit time (mTTI), the required mean time for contrast microbubbles to transit the ROI; time to peak (TTP), the time elapsed between the begging of the examination to highest intensity peak; wash-in rate (WiR); wash-in perfusion index (WiPI); wash-out area under the curve (WoAUC); wash-in area under the curve (WiAUC) and wash-in and wash-out area under the curve (WiWoAUC); fall time (FT); and wash-out rate (WoR) (Figure 3). The quality of this process was determined using the software quality of fit parameter (QOF), whose value was considered appropriate if the 50% threshold was exceeded.

### 2.4. Histopathology and Immunohistochemical Analysis

Biopsy tissue fragments were fixed in neutral buffered formalin, routinely processed for paraffin embedding, and 4 µm seriate sections were cut. Hematoxylin and eosin stained slides were utilized to ascertain the diagnosis, and two consecutive sections were further utilized for immunostaining blood vessels with either anti-CD105 or anti-CD31 antibodies. Briefly, after antigen retrieval in 0.1 M citrate buffer pH6 for 20 min, the sections were incubated in a 1% hydrogen peroxide solution for 30 min to block the endogenous peroxidase activity and then kept for another 30 min in 3% skimmed milk in PBS (Phosphate Buffer Saline) for blocking unspecific antigen sites. The primary antibodies were incubated on the slides at 4 °C for 18 h (mouse anti-CD31, 1:50, Dako, Glostrup, Denmark) or (rabbit anti-CD105, 1:100, Thermo Fisher Scientific, Waltham, MA, USA), and the next day, the signal was amplified for 60 min utilizing a species-specific peroxidase polymer-based system (Vector Laboratories, Burlingame, CA, USA). The signal was detected with 3,3′-diaminobenzidine (DAB) (Vector Laboratories), and the slides were coverslipped in a xylene-based mounting medium (Sigma–Aldrich, St. Louis, MO, USA) after a hematoxylin counterstaining. For each of the two antibodies, all slides were processed at the same time for protocol consistency and semi-quantitative purposes.

After immunostaining, all the slides were scanned with a ×20 objective using a Motic EasyScan Pro6 slide scanner (Motic Europe, Barcelona, Spain), the resolution being sufficient to manually define the maximum diameter (vd) of each immunopositive vessel as well as the area of each tissue fragment. The total vascular area per square millimeter (MVD) of the tissue fragments was thus determined and further utilized for comparative analysis. All image measurements were performed using the Motic DSAssistent package.

### 2.5. Statistical Analysis

Statistical analyses were performed with SPSS v. 25.0 (SPSS Inc., Chicago, IL, USA). Data were represented as mean ± standard deviation (SD) or standard error of the means (SE), and median (interquartile range, IQR) when continuous variables were reported or the number of patients and percentages when categorical variables were reported. Kolmogorov–Smirnov test was used for checking normality, and in accordance with the results, the Mann–Whitney U test was applied for comparing the groups of patients (M0 vs. M1). Categorical variables were compared using χ2 test. All the tests where *p*-values ≤ 0.05 were considered statistically significant.

Spearman correlations were assessed to establish the relations between variables and visualized as a scatter matrix plot to look for the strength and directions of all correlations. According to missing data, the used procedure was a pairwise deletion of cases.

## 3. Results

Out of 56 patients diagnosed with gastric tumors, 6 patients were excluded due to another histologic type of cancer (5 patients were diagnosed with gastric lymphoma and one with GIST). For 16 patients, only the EUS TNM stage was assessed, and therefore, they were excluded due to lack of CHI-EUS examination. For the rest of the 34 patients, we performed a retrospective analysis of EUS TNM stages, CHI-EUS parameters after video processing with VueBox, immunohistochemical results of staining with CD105, and CD31. Consecutively, all data were examined for the remaining 34 patients, and a total of 80 ROI were correlated with the vd and MVD (Figure 4).

The most important clinical and pathological patient characteristics are described in Table 1.

After quantifying the number of vessels, it was clear that there were almost no CD105 positive vessels in control tissue (0.64 ± 0.40/mm^2^) compared to adenocarcinoma areas (25.33 ± 9.94/mm^2^), *p* = 0.010 (Figure 5A). CD31-positive vessels, on the other hand, could not distinguish between control (42.48 ± 8.19/mm^2^) and tumor areas (50.26 ± 13.69/mm^2^), with almost the same vascular densities for the two histopathological states (*p* = 0.33). As expected, in all instances, immunostaining for CD31 revealed more vessels than for CD105 for both control and tumor regions.

Regarding the average maximum diameters (Figure 5B), both CD31- and CD105-positive vessels showed clear-cut differences between control (19.50 ± 1.19 µm; 4.18 ± 2.63 µm) and adenocarcinoma areas (32.04 ± 2.84 µm; 30.94 ± 3.66 µm) (*p* < 0.001), with the widest difference being observed for CD105. The diameters were also reduced for CD105 vessels compared to CD31-positive vessels only for control tissue but with no differences for the adenocarcinoma tissue.

Altogether, vd and MVD analysis revealed that CD105-positive vessels showed an abrupt increase in density and diameters from control to adenocarcinoma, with both CD31-positive and 105-positive vessel diameters increasing in cancer tissue, probably to support higher metabolic rates and energy requirements in cancer. Moreover, tumor transformation homogenizes the morphology of total and angiogenic vasculature, with no differences between CD31- and CD105-labeled vessels in adenocarcinoma tissue.

As in Spearman’s correlation coefficient from Table 2, positive strong correlations were found between tumor grade and seven CHI-EUS parameters, namely PE, WiAUC, WiR, WiPI, WoAUC, WiWoAUC, and WoR. The 95% confidence bands from the simple regressions in Figure 6 give a visual sense of how strongly the parameters are correlated and if there is a positive or negative relationship.

We roughly determined the correlations between our variables; the boxes on the lower left-hand side of the whole scatterplot are in mirror images of the plots on the upper right hand. If the plot looks like a line, as in WiAUC and PE, it is safe to say that there is a positive correlation between them; otherwise, more statistical analysis would be needed to verify this correlation, as in Table 2.

The degree associations between two parameters were assessed and correlations were found between vd and MVD and CHI-EUS parameters (Table 2 and Figure 6). Large correlations with high statistical significance were found between vd CD31 and PE, WiAUC, WiR, WiPI, FT, and WoR. Negative correlations with statistical significance were also found between MVD CD105 and PE, WiAUC, WiR, WiPI, WoAUC, WiWoAUC, and WoR and between MVD CD31 and PE and WiPI.

Differences in perfusion pattern for M0 versus M1 patients were also tested in this study. MVD CD105 and RT were two parameters that have statistical difference values between metastatic status and M0 GC (higher values for MVD CD105 in M0 than in M1, *p* = 0.002, respectively; smaller values for RT in M0 than in M1, *p* = 0.022) (Table 3). Quality of fit was found to be significantly higher for M0 than for M1 (*p* = 0.022).

## 4. Discussion

GC angiogenesis assessment may play a pivotal role in the oncologic management of the disease. With few clinical trials already underway on targeting angiogenesis, new diagnosis and prognostic opportunities are still required to improve patient survival [17,18,19]. Our study objective was to test the feasibility of CHI-EUS on GC by comparing it with available angiogenesis immunohistochemical staining markers. Shifting the angiogenesis process to a non-invasive assessment by considering a time-intensity curve of CHI-EUS seems feasible and may aid in the diagnosis and prognostic purposes.

While this concept is not new, this is the first time it is considered for GC. While tumor angiogenesis is a major factor in the cancer evolution process, it is currently assessed by considering the microvessel density on a biopsy or tissue specimen. However, this might not be sufficient for GC angiogenesis assessment. Tumor invasiveness as well as the fact that only a tumor piece is examined might hamper the actual results. Therefore, non-invasive imaging examinations such as CHI-EUS could cover the entire tumor and have the advantage of potentially providing real-time in vivo assessment, which may provide more details about the patient’s prognosis.

To our knowledge, this is the first study that assesses GC tumor perfusion in a real-time situation by using CHI-EUS and comparing it to known angiogenesis IHC markers. Imaging angiogenesis has been considered before in different settings to highlight a perfusion pattern, thus differentiating between benign and malignant tumors. One of the challenging aspects when considering GC’s real-time perfusion assessment is the peristaltic movement that may hamper the image acquisition process. However, along with fast imaging techniques, perfusion CT (PCT) and dynamic contrast-enhanced DCE-MRI have correlated perfusion parameters with tumor stage and histologic grade [20,21,22]. Furthermore, these techniques may be used to assess the tumor response after oncologic therapies. Consecutively, transabdominal contrast enhanced-ultrasound was also considered for GC diagnosis with morphologic patterns and enhancement patterns [23]. CHI-EUS seems to be a more attractive method, first of all, because of the safety profile and the lack of ionization process, the fact it may be used for patients with renal dysfunction, as well as the respiratory motion correction techniques in free-breathing CHI-EUS. However, a EUS setting offers more advantages than transabdominal ultrasound since direct contact with the tumor will be performed, and bowel gas may be easily obviated [24,25,26]. Moreover, not only tumors located in the lower part of the stomach may be indicated but also the ones located in the upper part of the stomach. However, it should be performed by an experienced endoscopist, as in the grading process, some flaws may appear for both upstaging and downstaging.

In our study, we compared the perfusion parameters provided by CHI-EUS for gastric tumors and found a positive correlation between the TIC parameters and MVD for both CD31 and CD105. Endoglin has been recognized as a potential predictor of hematogenous recurrence in GC, thus providing a possible relationship between newly formed vessels and locoregional development [27,28].

Although surface mucosal biopsies reveal only the superficial inter-crypt chorion, without reflecting the changes in the deeper lamina propria, our study proves that even these small amounts of interglandular connective tissue may show significant differences when comparing control and adenocarcinoma areas. Our results showed a higher level of CD105 within the tumor samples than in normal tissue, highlighting tumor vessels. CD105 has a higher expression in vascular endothelial cells, mostly at the tumor edge, suggesting its spread potential. Moreover, it was found useful as a potential predictor of GC recurrence after surgery, which may suggest a relationship with hematogenous recurrence [29]. We also measured the vessel density by using the platelet endothelial cell adhesion molecule CD31, which was found to have a major role in the tumor microenvironment vessels. CD31 was found responsible for the formation of vasculogenic mimicry channels, which allow the formation of vascular channels transporting fluid from leaky vessels or connecting with normal blood vessels [30].

We also tried to correlate the TIC parameters with factors related to tumor prognoses such as tumor grade, tumor size, and metastasis. Traditionally, a high MVD within the tumor might indicate a poorer outcome and may relate to a more aggressive tumor [31,32]. Nowadays, pathology assessment may be performed by whole-in slide imaging techniques; however, vascular spots may still not cover the slide, and a proper evaluation might be difficult [33]. On the other hand, CHI-EUS could provide a real-time vascular evaluation of the tumor. It is also noteworthy that this is a repeatable technique and could be a more reliable method for tumor blood flow and volume [32,34].

Intratumoral angiogenesis followed by TIC analysis by using commercial software available for ultrasound platforms makes our results reproducible. Moreover, the endosonographer has the possibility to select the region of interest, which may seem more suitable for tumor vascularity and may maximize the use of EUS in this setting. However, this may require an additional step in the diagnosis process, as it is not directly incorporated into the ultrasound device.

The role of EUS in gastric tumors has evolved over the years. Traditionally, EUS is acknowledged by available guidelines as the main method to assess the T stage in gastric tumors and should be considered after pathologic diagnosis [35]. Further on, this technique may be also used when gastric outlet obstruction is present, secondary to antral GC, by performing a EUS gastrojejunal anastomosis using lumen-apposing metal stents [36,37]. This proof-of-concept study may impact routine EUS-imaging examination of GC, as it may be easily used in the local extension examination of a gastric tumor by using a contrast agent. However, it requires a more thorough examination of the gastric tumor, thus increasing the examination time.

EUS contrast examination might be a benchmark for microvasculature assessment, as it provides information on the contrast influx and washout within the tumor [38]. While its main field of application was to differentiate benign from malignant lesions of the pancreas [39], currently, it has also been used EUS-guided tissue acquisition performance in order to choose with more precision puncture area [40]. In addition, by using a contrast agent in a EUS setting after tumor radiofrequency ablation, intratumoral vessels might be highlighted, and residual tissue may be targeted in another session [41]. When discussing CHI-EUS, a similar study that focused on colonic cancer perfusion assessment suggested that longitudinal monitoring of antiangiogenic therapies may aid disease monitoring [42]. The authors emphasized the CHI-EUS might be considered technically demanding and would generally require advanced endoscopic skills, mainly because of peristalsis and tumor position, which may lead to unintentionally endoscopic movement.

Our approach is feasible, offers a new perspective for EUS in assessing GC, and may also be considered as a potential tool to assess treatment response for future oncologic therapies.

We acknowledge that the major limitation of our study is the small number of patients and the fact that even though we could assess the perfusion pattern after contrast injection, computer analysis was performed after uploading the films in the dedicated software. When discussing the technique, we did not assess the normal gastric wall by CHI-EUS because it is rather thin even though the layers are clearly visible in B mode. We also did not consider other pathologic situations that may provide a thicker gastric wall. Furthermore, we mostly focused on advanced tumor stages that might provide a heterogeneous group.

Nonetheless, by including other histologic types of gastric tumors, the selection criteria might be expanded, and differences between diffusion patterns may be encountered. Thus, we considered adenocarcinoma as a starting point for this proof-of-concept study, laying the grounds for future developments.

## 5. Conclusions

GC angiogenesis assessment by CHI-EUS is feasible and may be considered for future studies based on TIC analysis. Thus, patients’ prognoses may be influenced, as new as new first-line therapies may be selected. While neoformation vessels are among the first morphologic alterations, this method may aid for the early prediction of therapeutic response. While EUS may not be considered for GC restaging, CHI-EUS could help in highlighting anti-angiogenic changes. However, more patients should be included in future studies before methodologic standardization may be considered.

## Figures and Tables

**Figure 1 jpm-12-01020-f001:**
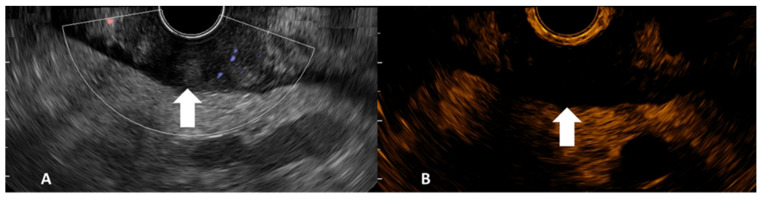
Gray-scale B-mode EUS image side-completed with Doppler mode (**A**) and contrast harmonic EUS image side (**B**) of gastric adenocarcinoma (tumor indicated by arrows).

**Figure 2 jpm-12-01020-f002:**
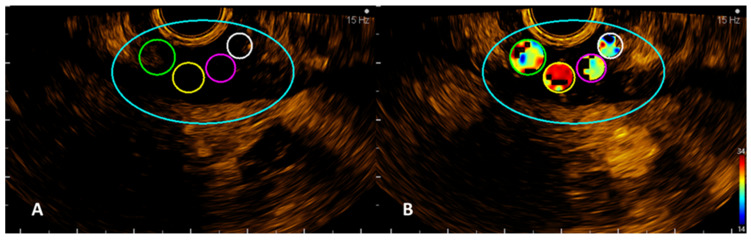
VueBox images with the four ROIs manually drawn (**A**) and an example of one of the automatically calculated parameters, rise time (RT) (**B**).

**Figure 3 jpm-12-01020-f003:**
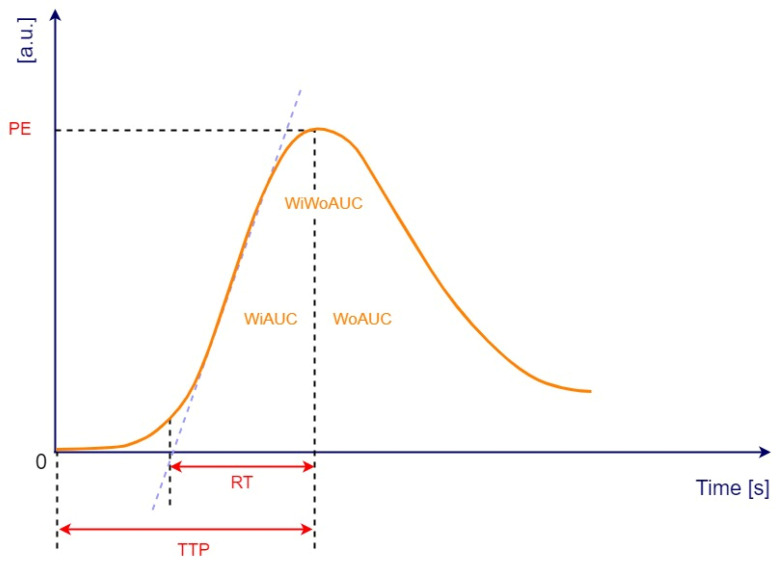
Time-intensity curve with following parameters: peak enhancement (PE), wash-in area under the curve (WiAUC), rise time (RT), time to peak (TTP), wash-out area under the curve (WoAUC), wash-in and wash-out area under the curve (WiWoAUC). a.u., arbitrary units; s, seconds. Adapted from VueBox^®^ Quantification Toolbox, Copyright© 2019 Bracco Suisse SA.

**Figure 4 jpm-12-01020-f004:**
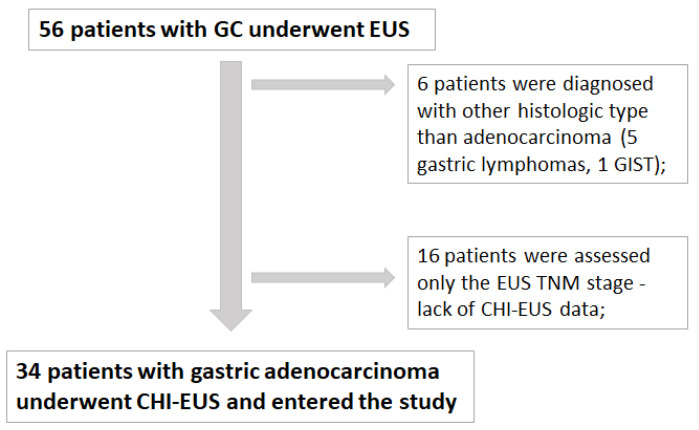
Total number of patients diagnosed with GC from 01.10.2019 to 01.01.2022 at Research Center of Gastroenterology and Hepatology of Craiova, University of Medicine and Pharmacy of Craiova, Romania.

**Figure 5 jpm-12-01020-f005:**
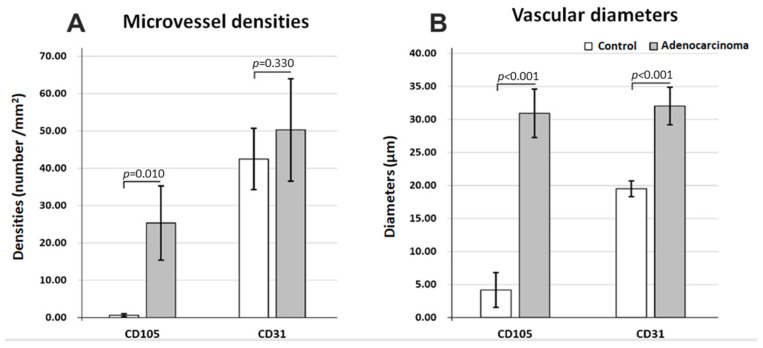
Microvasculature differences in normal (control) vs. adenocarcinoma tissue immunostaining for CD31 and CD105, average values for microvessel densities (**A**) and for maximum vascular diameters (**B**).

**Figure 6 jpm-12-01020-f006:**
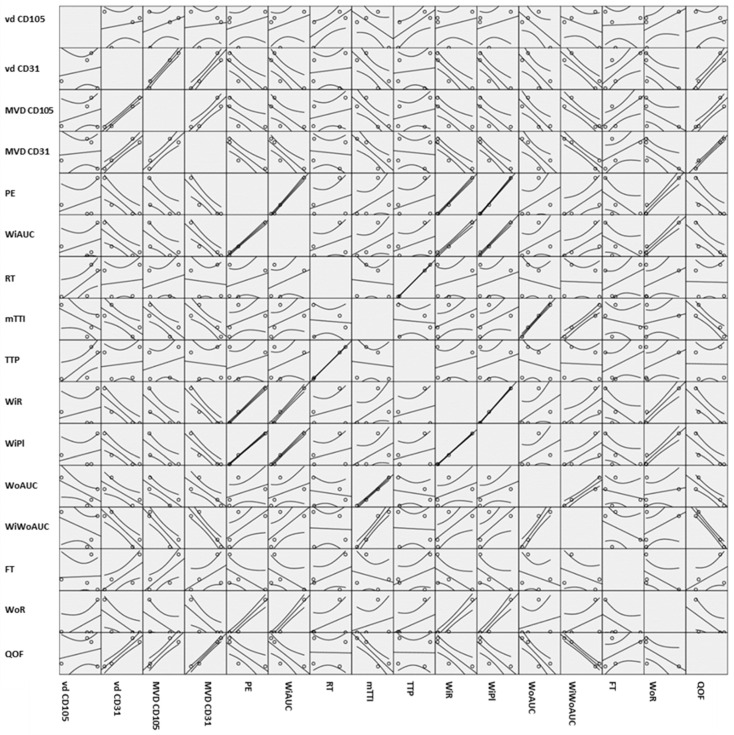
Scatter matrix plot containing all the pairwise scatter plots of the variables. vd CD105, vessel diameter reported for CD105; vd CD31, vessel diameter reported for CD31; MVD CD105, microvascular density reported for CD105; MVD CD31, microvascular density reported for CD31; PE, peak enhancement; WIAUC, wash-in area under the curve; RT, rise time; mTTI, mean transit time; TTP, time to peak; WIR, wash-in rate; WiPI, wash-in perfusion Index; WoAUC, wash-out AUC; WIWoAUC, wash-in and wash-out AUC; FT, fall time; WoR, wash-out rate; QoF, quality of fit.

**Table 1 jpm-12-01020-t001:** Patients’ characteristics for patients included in the study (*n* = 34).

Patients’ Characteristics	Patients (Total Number = 34) *n* (%)
Age-mean (range)	65.7 (43–86)
**Gender (male/female)**	24/10
**Tumor location**	
Cardia	0
Body	20 (59%)
Antrum	10 (29%)
Others	4 (12%)
Borrmann	
Polypoid	0
Fungating	0
Ulcerated	6 (18%)
Infiltrative	4 (12%)
UI	10 (29%)
UF	14 (41%)
Tumor differentiation	
G1	2 (6%)
G2	14 (41%)
G3	18 (53%)
T stage	
T1	2 (6%)
T2	2 (6%)
T3	10 (29%)
T4	20 (59%)
N stage	
N0	10 (29%)
N1	0
N2	12 (35%)
N3	12 (35%)
Dimensions	
<5	4 (12%)
5–10	14 (41%)
>10	16 (47%)

WHO, World Health Organization; T and N stages, assessed by EUS; n, number of patients for each characteristic; %, percentage of patients for each characteristic.

**Table 2 jpm-12-01020-t002:** Correlation coefficients between IHC and CHI-EUS parameters, size, and grade of the tumor.

Correlations																			
*ρ* (Spearman Coefficient)	vd CD105	vd CD31	MVD CD105	MVD CD31	PE	WiAUC	RT	mTTI	TTP	WiR	WiPl	WoAUC	WiWoAUC	FT	WoR	QOF	Area	Size	Grade
**vd CD105**	1.00																		
**vd CD31**	0.54 *	1.0040																	
**MVD CD105**	−0.147	0.176	1.00																
**MVD CD31**	−0.3	−0.06	0.53 *	1.00															
**PE**	0.07	−0.82 **	−0.75 **	−0.70 *	1.00														
**WiAUC**	0.04	−0.67 *	−0.82 **	−0.40	0.78 **	1.00													
**RT**	0.32	0.36	0.46	−0.10	−0.47 *	0.01	1.00												
**mTTI**	−0.32	−0.56	−0.14	0.10	−0.07	0.42	0.46 *	1.00											
**TTP**	0.32	0.356	0.50	−0.10	−0.37	−0.08	0.89 **	0.20	1.00										
**WiR**	0.04	−0.98 **	−0.79 **	−0.20	0.95 **	0.72 **	−0.60 **	−0.09	−0.53 *	1.00									
**WiPl**	0.07	−0.82 **	−0.75 **	−0.70 *	1.00 **	0.78 **	−0.47 *	−0.07	−0.37	0.95 **	1.00								
**WoAUC**	−0.04	−0.56	−0.68 **	−0.30	0.48 *	0.62 **	0.22	0.37	0.15	0.26	0.48 *	1.00							
**WiWoAUC**	−0.04	−0.56	−0.68 **	−0.30	0.56 **	0.81 **	0.09	0.46 *	−0.10	0.43	0.56 **	0.82 **	1.00						
**FT**	0.00	0.82 **	0.42	0.00	−0.56 **	−0.26	0.59 **	0.31	0.46 *	−0.75 **	−0.56 **	0.37	0.18	1.00					
**WoR**	−0.11	−0.98 **	−0.75 **	−0.60	0.93 **	0.73 **	−0.54 *	−0.13	−0.46 *	0.96 **	0.93 **	0.18	0.39	−0.77 **	1.00				
**QOF**	−0.43	0.67 *	0.57 *	0.00	−0.26	−0.48 *	−0.13	−0.61 **	0.02	−0.31	−0.26	−0.42	−0.53 *	0.14	−0.18	1.00			
**Area**	−0.85 **	−0.26	0.13	−0.62	0.09	0.25	−0.1	0.3	−0.03	0.07	0.09	−0.14	0.06	−0.02	0.23	0.25	1.00		
**Size**	0.32	0.41	−0.38 *	−0.45 *	0.13	0.20	−0.20	−0.16	−0.09	0.13	0.13	−0.20	−0.02	−0.05	0.29	0.57 **	0.79 **	1.00	
**Grade**	0.25	−0.21	−0.47 *	−0.23	0.78 **	0.50 *	−0.42	−0.21	−0.42	0.66 **	0.78 **	0.60 **	0.51 *	−0.24	0.63 **	0.01	−0.15	0.15	1.00

Spearman test r values, *, *p*-value < 0.05; **, *p*-value < 0.01. vd CD105, vessel diameter reported for CD105; vd CD31, vessel diameter reported for CD31; MVD CD105, microvascular density reported for CD105; MVD CD31, microvascular density reported for CD31; PE, peak enhancement; WIAUC, wash-in area under the curve; RT, rise time; mTTI, mean transit time; TTP, time to peak; WIR, wash-in rate; WiPI, wash-in perfusion index; WoAUC, wash-out AUC; WIWoAUC, wash-in and wash-out AUC; FT, fall time; WoR, wash-out rate; QoF, quality of fit; Area, ROI area; Size, tumor size; Grade, tumor grade.

**Table 3 jpm-12-01020-t003:** Differences between IHC and CHI-EUS parameters, size, and tumor grade in M0 vs. M1 GC.

Missing Value Number/Mean ± SD Median (IQR)	Total (*N* = 34)	M0 (*N* = 24)	M1 (*N* = 10)	M0 vs. M1 *p*-Value
vd CD105	6/31.27 ± 12.53	6/29.61 ± 14.4	0/34.25 ± 7.63	0.109
	30.61 (24.61–38.69)	26.05 (16.47–36.34)	30.75 (29.19–41.76)	
vd CD31	12/31.72 ± 10.43	10/32.96 ± 12.47	2/29.56 ± 5.41	0.815
	27.15 (25.48–34.96)	27.15 (25.48–34.96)	28.07 (24.89–35.72)	
MVD CD105	6/23.97 ± 9.52	6/31.69 ± 39.76	0/10.07 ± 13.65	0.002 **
	9.52 (5.36–26.56)	15.04 (7.53–34.52)	4.75 (2.78–12.97)	
MVD CD31	12/54.26 ± 48.50	10/58.28 ± 58.41	2/47.22 ± 25.14	0.815
	46.73 (20.73–85.39)	46.73 (12.07–93.03)	38.52 (27.15–75.99)	
PE	14/8238.95 ± 21,033.93	8/9412.13 ± 23,454.99	6/3546.26 ± 3826.89	0.750
	881.52 (232.07–1849.57)	881.52 (230.22–1763.49)	3546.26 (232.07–6860.45)	
WiAUC	14/6040.21 ± 6232.43	8/4379.4 ± 2923.04	6/12,683.44 ± 11,389.9	0.494
	3724.65 (2819.43–6655.4)	3724.65 (2638.09–6033.93)	12,683.44 (2819.43–22,547.45)	
RT	14/12.24 ± 9.63	8/10.99 ± 10.44	6/17.21 ± 1.33	0.022 *
	9.94 (5.85–16.06)	8.57 (5.64–11.12)	17.21 (16.06–18.36)	
Mtti	14/127.91 ± 85.87	8/116.89 ± 91.04	6/172.01 ± 44.92	0.148
	99.31 (71.39–210.9)	79.76 (54.38–208.52)	172.01 (133.11–210.9)	
TTP	14/17.16 ± 11.31	8/15.96 ± 12.4	6/21.991 ± 1.69	0.064
	14.67 (8.19–21.62)	11.95 (7.38–20.04)	21.99 (20.53–23.45)	
WiR	14/109,797.92 ± 335,894.79	8/136,779.39 ± 372,865.61	6/1875.99 ± 2138.78	1.00
	158.09 (61.1–1221.71)	158.09 (61.36–1064.91)	1875.99 (23.75–3728.22)	
Wipl	14/4702.92 ± 11,684.22	8/5332.02 ± 13,027.51	6/2186.51 ± 2344.99	0.750
	588.54 (155.69–1335.99)	588.54 (151.63–1251.66)	2186.51 (155.69–4217.33)	
WoAUC	14/13,179.78 ± 166,866.63	8/13,139.21 ± 18,548.46	6/13,342.05 ± 6573.04	0.148
	5649.26 (2123.69–19,034.47)	4471.23 (1675.12–25,024.78)	13,342.05 (7649.63–19,034.47)	
WiWoAUC	14/19,879.21 ± 19,688.08	8/19,301.17 ± 21,132.76	6/22,191.39 ± 14,596.28	0.290
	9538.27 (7613.77–34,832.14)	8642.89 (4409.96–34,571.79)	22,191.39 (9550.64–34,832.14)	
FT	14/18.66 ± 9.71	8/17.14 ± 7.41	6/24.76 ± 16.14	0.750
	17.67 (13.12–21.53)	17.67 (13.88–21.24)	24.76 (10.78–38.74)	
WoR	14/106,145.06 ± 324,480.34	8/131,918.83 ± 360,303.79	6/3049.99 ± 3514.07	0.750
	51.47 (19.24–144.5)	51.46 (19.41–141.42)	3049.99 (6.71–6093.26)	
QoF	14/45.96 ± 19.27	8/49.78 ± 19.54	6/30.71 ± 7.49	0.022 *
	38.8 (30.36–70.85)	42.24 (32.36–72.05)	30.71 (24.22–37.2)	
AREA	14/0.26 ± 0.14	8/0.28 ± 0.15	6/0.18 ± 0.0	0.022 *
	0.2 (0.18–0.3)	0.25 (0.2–0.31)	0.18 (0.18–0.18)	
Size	7.71 ± 2.62	7.58 ± 2.69	8 ± 2.58	0.615
	8 (5–10)	8 (5.25–10)	10 (5–10)	
GRADE				0.466
1	2 (5.9%)	2 (8.3%)	0	
2	14 (41.2%)	10 (41.7%)	4 (40%)	
3	18 (52.9%)	12 (50%)	6 (60%)	

*, *p*-value < 0.05; **, *p*-value < 0.01. Mann–Whitney U test. vd CD105, vessel diameter reported for CD105; vd CD31, vessel diameter reported for CD31; MVD CD105, microvascular density reported for CD105; MVD CD31, microvascular density reported for CD31; PE, peak enhancement; WIAUC, wash-in area under the curve; RT, rise time; mTTI, mean transit time; TTP, time to peak; WIR, wash-in rate; WiPI, wash-in perfusion index; WoAUC, wash-out AUC; WIWoAUC, wash-in and wash-out AUC; FT, fall time; WoR, wash-out rate; QoF, quality of fit; Area, ROI area; Size, tumor size.

## Data Availability

All the data presented in this study are available upon reasonable request from the corresponding author.

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
