# Peer review of "Gastric Cancer Angiogenesis Assessment by Dynamic Contrast Harmonic Imaging Endoscopic Ultrasound (CHI-EUS) and Immunohistochemical Analysis—A Feasibility Study"

_jpm, 2022, doi:10.3390/jpm12071020_

Round 1
Reviewer 1 Report
The study entitled “Gastric Cancer Angiogenesis Assessment by Dynamic Contrast 2 Harmonic Imaging Endoscopic Ultrasound (CHI-EUS) and Immunohistochemical Analysis – a feasibility study “ is reviewed. The authors aimed to test the feasibility of CHI-EUS on GC by comparing it with available angiogenesis immunohistochemical staining markers. They studied gastric adenocancer patients and reported that GC angiogenesis assessment by CHI-EUS is feasible and may be considered for future studies based on TIC analysis.
As a clinician, I found the topic quite interesting. Although I think that it would be better if other subtypes of stomach cancer were also included in the study, I did not criticize about this issue, as the authors indicated in the discussion that they would detail their work on this issue later. I think that the study opens a way for new studies, especially related to its use in predicting treatment response.
It is an interesting study that will contribute to the literature.
Author Response
We appreciate your comments, and we will continue our work as this was just a feasibility study
Reviewer 2 Report
Notes on the manuscript:
There are some spelling mistakes please revise the manuscript.
Title: the title mentioned (a feasible study), but it is not clear whether this is an artificial intelligence study with stored videos or a clinical study. I recommend that these details are mentioned in the title instead of the term feasibility study.
The abstract:
The type of clinical study and the total number of patients are not mentioned, and the number of videos mentioned in the results, kindly state both in the methodology.
The introduction: the introduction is not focused on the topic and the studies conducted in the same area before this study, but delves into molecular markers. Also, the part mentioning the abdominal CEUS seems out of topic, better to mention CE-EUS and its role in the diagnosis of various GIT tumors.
The methodology:
The mention of ((2) failure to provide informed consent) in the exclusion and early in the inclusion criteria is redundant; please don’t duplicate items in the inclusion and exclusion criteria.
The results:
1- The authors stated “Out of 56 patients diagnosed with gastric tumors, 6 patients were excluded due to 178 another histologic type of cancer (5 patients were diagnosed with gastric lymphoma and 179 one with GIST). For 16 patients only the EUS TNM stage was assessed and therefore they 180 were excluded due to lack of CHI-EUS examination. For the rest of the 34 patients,” . Could the authors present their cases in a flowchart for better understanding.
2- Are the patients’ characteristics in table 1 for the 34 patients that were finally included or for the whole population before exclusion?. Please state in the table
3- Table 2 states all the correlations in the study, but it is in the raw statistical format which might be hard for readers to understand. Please state your table in a more readable format as the variables and the significant statistical levels as r and pvalue in the table.
4- The authors stated the exact statistical results in both a written format and a table format, please choose one r the other as this multiplication is redundant. Also in the results section the authors should state what is unique and significant in their results to attract the attention of the readers, not just state the P values of all the variables.
5- The statistical methods used are limited to correlation and p value, it would be of more statistical value if there was risk ratio or odd’s ratio or confidence interval, also the multivariate analysis and ROC curve is more suitable to this kind of study comparing two diagnostic procedures. Could the authors modify?. I know the authors used scatter matrix plot, but most reader may not be familiar with, could they explain in the manuscript how it is read and what is its significance in this study? And if offers any more data than the correlation stated earlier?
Discussion:
1- The fact that this is stored video recorded and then analyzed by AI could this affect the use as a live diagnostic procedure in clinical practice, could the authors state the limitations?
2- A large part of the discussion is mentioning EUS per se, which is out of topic, this is CE-EUS which is a different diagnostic measure all together, unless the discussion is about what the CE-EUS or CHI-EUS offers more than the EUS, please modify.
3- Please compare with the doppler results and if this method CHI-EUS offers more diagnostic assessment more than the regular doppler to justify the cost?
The conclusions:
The authors stated, “A major impact may be achieved on patient prognosis, 356 as the first-line therapies may be selected.” This could be an overstatement of a pilot study. Please modify.
Author Response
We are very grateful for the constructive comments from you. We also thank you for the time and effort in reviewing our manuscript. We have carefully addressed point-by-point all the comments.
There are some spelling mistakes please revise the manuscript.
Title: the title mentioned (a feasible study), but it is not clear whether this is an artificial intelligence study with stored videos or a clinical study. I recommend that these details are mentioned in the title instead of the term feasibility study.
We appreciate your suggestion. Our study is the first of its kind and actually, we consider it a feasibility study because our aim is to uncover both the opportunities and disadvantages that using our setting may be encountered.
Basically, the idea is to provide a new technique that may aid for future assessment of tumor angiogenesis in a real-time setting. Here are some recent studies published for other organs which focus on a similar idea. CEUS (PMID: 35284253 ), MRI contrast enhancement (PMID: 35369721), Dynamic triple phase enhanced CT perfusion imaging (PMID: 34717573).
Dynamic vascular pattern (DVP) is a function of quantification software designed for the evaluation of tissue perfusion obtained with real-time CEUS examination, which is true for the transcutaneous and also EUS approach.
The abstract:
The type of clinical study and the total number of patients are not mentioned, and the number of videos mentioned in the results, kindly state both in the methodology.
We modified the abstract according to your indications
The introduction: the introduction is not focused on the topic and the studies conducted in the same area before this study, but delves into molecular markers. Also, the part mentioning the abdominal CEUS seems out of topic, better to mention CE-EUS and its role in the diagnosis of various GIT tumors.
We appreciate your comment, however, our study objective, angiogenesis assessment by endoscopic ultrasound is a novel technique and has not been proposed so far for gastric cancer. The use of molecular markers for angiogenesis is rather important as they lay the ground for our feasibility study because we compared the analyzed parameters with the vascular diameter and microvascular density. Thus, the presentation of the available immunohistochemical markers is rather relevant so that we may introduce our study.
On the other hand, abdominal CEUS is relevant since, as described above that are studies that used contrast enhancement for vascular pattern assessment, especially in hepatocellular carcinoma (PMID) or more recently PMID: 35284253 recently.
Contrast-enhanced Endoscopic Ultrasound is currently used for pancreatic tumor assessment to distinguish between PDAC, neuroendocrine tumors or chronic pancreatitis. Noteworthy is that so far when discussing gastric tumors, contrast-enhanced endoscopic ultrasound has been used especially for the differential diagnosis of the subepithelial tumors and more recently it has been proposed for lymph-node assessment. We inserted another paragraph in the introduction
The methodology:
The mention of ((2) failure to provide informed consent) in the exclusion and early in the inclusion criteria is redundant; please don’t duplicate items in the inclusion and exclusion criteria.
We made the required changes.
The results:
- The authors stated “Out of 56 patients diagnosed with gastric tumors, 6 patients were excluded due to 178 another histologic type of cancer (5 patients were diagnosed with gastric lymphoma and 179 one with GIST). For 16 patients only the EUS TNM stage was assessed and therefore they 180 were excluded due to lack of CHI-EUS examination. For the rest of the 34 patients,” . Could the authors present their cases in a flowchart for better understanding.
We agree with your observation and we included a flowchart.
2- Are the patients’ characteristics in table 1 for the 34 patients that were finally included or for the whole population before exclusion?. Please state in the table
As recommended we modified the table’s description and also in the table
3- Table 2 states all the correlations in the study, but it is in the raw statistical format which might be hard for readers to understand. Please state your table in a more readable format as the variables and the significant statistical levels as r and pvalue in the table.
You are right about making Table 2 easier to understand and we presented the r values using only 2 digits. We consider including p-values in the table will burden the table with information and make it difficult to read. The p-values are declared using * for p-value < 0.05 and ** for p-value < 0.01 (we did not obtained p-values less than 0.001 or 0.0001).
4- The authors stated the exact statistical results in both a written format and a table format, please choose one r the other as this multiplication is redundant. Also in the results section the authors should state what is unique and significant in their results to attract the attention of the readers, not just state the P values of all the variables.
Deleting the information about r and p value in the lines 249-253, 796-801 helps us underline better our results regarding the positive or negative correlations between the assessed parameters. These results are covered in the Discussion section. Thank you for your suggestion.
5- The statistical methods used are limited to correlation and p value, it would be of more statistical value if there was risk ratio or odd’s ratio or confidence interval, also the multivariate analysis and ROC curve is more suitable to this kind of study comparing two diagnostic procedures. Could the authors modify?. I know the authors used scatter matrix plot, but most reader may not be familiar with, could they explain in the manuscript how it is read and what is its significance in this study? And if offers any more data than the correlation stated earlier?
We explained how the scatter matrix plot is read in the lines 175, 228-229. We added more information for an easier understanding.
We roughly determined the correlations between our variables, the boxes on the lower left-hand side of the whole scatterplot are in mirror images of the plots on the upper right hand. If the plot looks like a line, as in WiAUC and PE, it is safe to say that there is a positive correlation between them, otherwise, more statistical analysis would be needed to verify this correlation, as in Table 2.
Thank you for your advice.
Discussion:
- The fact that this is stored video recorded and then analyzed by AI could this affect the use as a live diagnostic procedure in clinical practice, could the authors state the limitations?
We agree with your observation and we mentioned this aspect in the Limitation paragraph from the discussions.
- A large part of the discussion is mentioning EUS per se, which is out of topic, this is CE-EUS which is a different diagnostic measure altogether, unless the discussion is about what the CE-EUS or CHI-EUS offers more than the EUS, please modify.
We agree with your observation and we included another paragraph within the discussion section
EUS contrast examination might be a benchmark for microvasculature assessment, as it provides information on the contrast influx and washout within the tumor. While its main field of application was to differentiate benign from malignant lesions of the pancreas, currently it has also been used EUS-guided tissue acquisition performance in order to choose with more precision puncture area. Also, by using a contrast agent in a EUS setting after tumor radiofrequency ablation, intratumoral vessels might be highlighted and residual tissue may be targeted in another session. When discussing CHI-EUS A similar study that focused on colonic cancer perfusion assessment suggested that longitudinal monitoring of antiangiogenic therapies may aid disease monitoring. The authors emphasized the CHI-EUS might be considered technically demanding and would generally require advanced endoscopic skills, mainly because of peristalsis and tumor position which may lead to unintentioanally endoscopic movement.
- Please compare with the doppler results and if this method CHI-EUS offers more diagnostic assessment more than the regular doppler to justify the cost?
Doppler techniques are usually associated with various artifacts, especially “flash” artifacts (caused by cardiac / respiratory movements) and “overflow” artifacts (especially after contrast injection). Furthermore, Doppler techniques are not useful for visualization of low-volume, low-velocity microvascular flow. Low mechanical index contrast harmonic imaging is devoid of artifacts and allows quantification using specific software like Vuebox. Also, the ultrasound contrast agent (Sonovue) is a blood pool agent which stays in the macro- and microvessels, with certain dynamic consequences as opposed to dynamic contrast and magnetic resonance imaging methods.
The conclusions:
The authors stated, “A major impact may be achieved on patient prognosis, 356 as the first-line therapies may be selected.” This could be an overstatement of a pilot study. Please modify.
We changed the phrase according to your recommendation.
Round 2
Reviewer 2 Report
I would like to thank the authors for their modifications.